# Evaluation of Parameters Affecting *Agrobacterium*-Mediated Transient Gene Expression in Industrial Hemp (*Cannabis sativa* L.)

**DOI:** 10.3390/plants13050664

**Published:** 2024-02-28

**Authors:** Tasnim Mohammad, Rishikesh Ghogare, Lauren B. Morton, Amit Dhingra, Shobha Potlakayala, Sairam Rudrabhatla, Sarwan K. Dhir

**Affiliations:** 1Center for Biotechnology, Department of Agricultural Sciences, Fort Valley State University, 113, Alva Tabor Building, Fort Valley, GA 31030, USA; tasnim.mohammad@fvsu.edu (T.M.); lmorton@wildcat.fvsu.edu (L.B.M.); 2Department of Horticultural Sciences, Texas A&M University, College Station, TX 77843, USA; rishikesh.ghogare@ag.tamu.edu (R.G.); adhingra@tamu.edu (A.D.); 3The Central Pennsylvania Research and Teaching Laboratory for Biofuels, Penn State Harrisburg, Middletown, PA 17057, USA; sdp13@psu.edu (S.P.); svr11@psu.edu (S.R.)

**Keywords:** *Cannabis*, *Agrobacterium tumefaciens*, GUS, GFP, transient, genetic transformation, hemp

## Abstract

Industrial hemp *Cannabis sativa* L. is an economically important crop mostly grown for its fiber, oil, and seeds. Due to its increasing applications in the pharmaceutical industry and a lack of knowledge of gene functions in cannabinoid biosynthesis pathways, developing an efficient transformation platform for the genetic engineering of industrial hemp has become necessary to enable functional genomic and industrial application studies. A critical step in the development of *Agrobacterium tumefaciens*-mediated transformation in the hemp genus is the establishment of optimal conditions for T-DNA gene delivery into different explants from which whole plantlets can be regenerated. As a first step in the development of a successful *Agrobacterium tumefaciens*-mediated transformation method for hemp gene editing, the factors influencing the successful T-DNA integration and expression (as measured by transient *β-glucuronidase* (GUS) and Green Florescent Protein (GFP) expression) were investigated. In this study, the parameters for an agroinfiltration system in hemp, which applies to the stable transformation method, were optimized. In the present study, we tested different explants, such as 1- to 3-week-old leaves, cotyledons, hypocotyls, root segments, nodal parts, and 2- to 3-week-old leaf-derived calli. We observed that the 3-week-old leaves were the best explant for transient gene expression. Fully expanded 2- to 3-week-old leaf explants, in combination with 30 min of immersion time, 60 µM silver nitrate, 0.5 µM calcium chloride, 150 µM natural phenolic compound acetosyringone, and a bacterial density of OD_600nm_ = 0.4 resulted in the highest GUS and GFP expression. The improved method of genetic transformation established in the present study will be useful for the introduction of foreign genes of interest, using the latest technologies such as genome editing, and studying gene functions that regulate secondary metabolites in hemp.

## 1. Introduction

Industrial hemp (*Cannabis sativa* L.) is a dioecious plant belonging to the Cannabinaceae family [1]. *Cannabis* is diploid and highly heterogenic, making traditional breeding costly and time-consuming. It is mainly grown for its fiber, seeds, oil, and medicinal compounds. Although *Cannabis* is known for its psychoactive properties due to its high level of delta-9 tetrahydrocannabinol (THC), hemp has a very low amount of THC (0.3% on a dry weight basis) and mainly produces cannabidiol (CBD) [2]. Due to its economic value and agronomical properties, hemp cultivation is on the rise in several parts of the United States and the CBD-related pharmaceutical market is one of the fastest growing in the world. 

Apart from the two major compounds, THC and CBD, the plant produces more than 100 additional phytocannabinoids in small quantities [3]. These compounds are believed to have therapeutic potential for treating conditions such as mood disorders, neurological disorders, cancer, diabetes, and pain [3,4]. The current knowledge of the phytocannabinoids’ biosynthetic pathways remains limited [5,6]. In addition, the introduction of novel traits is difficult to accomplish without changing the cannabinoid profile. Therefore, it is necessary to develop a replicable high-efficiency protocol to introduce genes into tissues and to regenerate plants from transformed cells of hemp. 

Biotechnological approaches, such as gene manipulation, could be an important landmark in hemp breeding, as demonstrated in other plant species, through the development of improved varieties that are resistant to biotic and abiotic stresses, with better nutritional and processing qualities [7,8,9]. The same approach can help further our understanding of the gene functions identified in transcriptomic, genomic, and proteomic studies in the cannabinoid pathways [7,8,9,10]. However, before the implementation of this technique in hemp species, it is imperative to identify totipotent cells using different explants to develop an efficient genetic transformation protocol that allows for the regeneration of transgenic plants in a short duration.

Transient gene expression is the temporary expression of genes after nucleic acids have been delivered into plant cells. This approach offers the advantages of being less affected by gene position and silencing effects, having a lack of heritable offspring for high biosafety, and the ability to analyze protein subcellular localization [11]. Multiple technologies have been utilized for introducing nucleic acids encoding an expression cassette into plant cells, e.g., the transformation of protoplasts, particle bombardment, and *Agrobacterium*-mediated transformation [12].

Among these, *Agrobacterium*-mediated transient transformation has significant advantages for broad deployment because it is a safe, high-level, and rapid technique to obtain a transient and high-level expression of target genes in a host plant cell nucleus. The technique is facile and versatile and has been used for the characterization of gene function in plants, including gene promoter properties, transcription factor activity, protein subcellular localization, and protein–protein interactions [13,14]. 

In recent years, transient expression systems using *Agrobacterium* have ceased to be exclusive to model plants, such as *Arabidopsis*, and individual transient expression systems for many economically important plants, such as *Brassica*, *Casella*, Solanum, *Capsicum annuum*, and tobacco have been established [15]. However, compared to these economic crops, the establishment of transient expression systems in medicinal plants is very much behind, which may leave the functions of genes incomplete and hinder the development and utilization of these natural medicinal values by humans. Since hemp is an important medicinal plant with great potential, it is essential to establish an efficient transient expression system for the analysis of phytocannabinoid synthesis-related gene functions.

It has been reported that *Cannabis* is recalcitrant to genetic transformation [16,17]; this might be due to the plant’s ability to protect itself against pathogen attacks, as the addition of ascorbic acid to scavenge excess reactive oxygen species (ROS) had a positive effect on the transformation efficiency [2]. To overcome this, the transient expression of GFP in *Cannabis* leaf mesophyll protoplasts has recently been reported [18,19,20,21].

Using *Agrobacterium*-mediated transformation, the first successful hemp transformation studies were reported by Feeney and Punja [22], followed by Wahby et al. [23] who showed that both hemp and drug-type *Cannabis* are susceptible to wild-type *A. tumefaciens* infection and it is thus not surprising that the most frequently used method for transient transformation is agroinfiltration by vacuum or by using a needleless syringe. A patent by Roscow [24] describes transient methods to transform *Cannabis* by vacuum infiltration or by dipping the green parts into an *Agrobacterium* soup after a vacuum is applied. The treated plant parts were trichomes on non-flowering parts of the plant that produce secondary compounds. It is, however, unclear if this property was inherited by the next generation.

A recent study explored the factors for optimizing transient expression, which included the hemp cultivars, *Agrobacterium* strain, chemical additives, physical aides (vacuum and sonication), and the hemp tissues and organs [2]. It was identified that adding Silwet-77, strain GV-3101, and a combination of vacuum and sonication improved transient GUS gene expression in leaf discs. The recent adaption of this protocol to transiently target *cannabinoid* biosynthetic genes to female inflorescences resulted in the modification of transcript levels and cannabinoid accumulation [25]. In addition, nanoparticle-based transient gene transformation of trichomes and leaf cells from one hemp variety, in which the transcription of soybean genes and localization of fluorescent-tagged transcription factor proteins were detected, has also been achieved [26].

Recently, a study reported the optimization of *Agrobacterium*-mediated transformation by using different tissue types, selection concentrations, and media types for hemp. The study identified hypocotyls from 7-day-old seedlings as an ideal explant with a selection pressure of 100 mg L^−1^ kanamycin. However, the transformation efficiency varied widely from 0 to 28.6% depending on the variety [27]. In another study, researchers expressed growth regulators, developmental regulators, and transcription factors like *WUSCHEL (WUS), SHOOT MERISTEMLESS (STM)*, *ISOPENTENYL TRANSFERASE (IPT)*, *GROWTH-REGULATING FACTOR (GRF)*, and *GRF-INTERACTING FACTOR (GIF)* to improve the regeneration rate for transformants. However, less than 3% of the hemp calli developed shoots [28,29]. However, many variables that can influence transformation frequencies following *Agrobacterium* infection such as the reporter gene, immersion time, explant type, genotype of the host plant, co-cultivation time, bacterial concentration, etc., remain to be explored. To further improve the transient expression method, our study tested factors such as the age of the leaves (1- to 3-week-old leaves and 2- to 3-week-old leaf-derived calli), type of explants (cotyledon, hypocotyl, root segment, and nodal part), calcium chloride/silver nitrate concentration, bacterial density, wounding method, and acetosyringone concentration for both GUS and GFP expression as well as other factors that are described elsewhere [30].

To the best of our knowledge, this article constitutes the first report to assess the aforementioned factors for the efficient transient expression of two marker genes in different types of tissues in *Cannabis sativa* L. The data obtained from previous research in combination with this study will lay the foundation for developing replicable and high-efficiency genetic engineering strategies in different hemp cultivars.

## 2. Results

### 2.1. Transient Gene Expression Efficiency Based on Different Types and Age of Explants

To test and optimize the agroinfiltration system, a wide range of hemp tissues were tested using GUS and GFP as reporter genes. Successful transient GUS expression was observed in all the tested tissue types: leaves, cotyledons, hypocotyls, root segments, nodal parts, and 2- to 3-week-old callus cultures (Figure 1A–J). Among all the tested tissue types, 3-week-old callus explants showed the highest transient GUS expression percentage (81.56%) with an average of 49.22 blue spots per explant (Table 1). At the same time, transient GFP fluorescence was detected in leaves, petioles, root segments, and callus explants (Table 2). In comparison, 3-week-old fully expanded wounded leaves explants had the highest percentage (59.44%) of GFP expression with an average of 166.44 GFP spots per explant compared to an average of 163.89 blue spots (Table 1 and Table 2). A cluster of multiple cells exhibiting GFP protein green fluorescence and chloroplast autofluorescence was captured using a Fluorescence Stereo Microscope under a GFP filter in the 3-week-old fully expanded leaves (Figure 2A–C). Strong expression of the GFP gene in primary mesophyll cells, including vascular structures, was observed 3 days after co-cultivation (Figure 2C–E). In the case of both GUS and GFP expression, 3-week-old leaf tissues with a surface area of 99.04 mm^2^ (Table 1 and Table 2 and Figure 3C) had the greatest number of blue or green fluorescence spots per explant (211.00 and 201.67, respectively). On the other hand, 1-week-old leaves with a surface area of 30.82 mm^2^ and 2-week-old leaves with a surface area of 55.85 mm^2^ had a lower number of blue foci and green fluorescence spots, as observed using a Stereo Zoom Microscope (Table 1 and Table 2 and Figure 3A,B). The lowest GUS expression was observed in hypocotyl explants (40.78%) while the lowest GFP expression was observed in root tissues (41.22%) (Table 1 and Table 2 and Figure 1F).

The transient expression frequency for GUS and GFP was generally higher in 3-week-old fully expanded leaves compared to 1-week-old and 2-week-old leaves (Table 1 and Table 2 and Figure 3A–C). One caveat of quantifying transient expression efficiency by counting the average number of spots per explant is that pretreatment of the explant tissue could influence the number of spots (Figure 2A–C). On average, 201 GFP-expressing cell clusters and 211 blue spots were observed.

### 2.2. Effect of Bacterial Cell Density on Transient Expression Efficiency

The bacterial density of the *Agrobacterium* culture during co-cultivation is an important factor that not only affects the transient transformation efficiency but also the regeneration potential of the explant. In this study, five different optical densities at OD_600nm_ (0.2, 0.4, 0.6, 0.8, and 1.0) were tested for GUS and GFP activity in the 3-week-old leaves. At OD_600nm_ = 0.4, the highest percentage of GUS- and GFP-expressing explants were observed (89.67% and 82.78%, respectively) and the highest number of GFP and blue spots (156.70 and 154.33, respectively) after 3 days of co-cultivation (Figure 4 and Figure 5). As the bacterial density increased in the co-cultivation media, the transformation efficiency decreased for both the GUS and GFP genes, with the lowest values observed at OD_600nm_ = 1.0 (Figure 4 and Figure 5). The appropriate bacterial density also depended on the type of explant used for the transformation; a higher density might cause an overgrowth of the bacteria, resulting in necrosis or ineffective recovery of the explants. At the same time, a lower optical density might result in insufficient bacterial cells to infect most of the cells, leading to a lower percentage of T-DNA transfer.

### 2.3. Effect of Wounding and Acetosyringone Concentration

In *Agrobacterium*-mediated transformation, wounding of the explant is necessary for efficient transformation. Generally, wounds provide *Agrobacterium* access to the plant cells. In this study, three different types of wounding were carried out using a surgical needle, a stainless steel metal screen, and sonication. The highest GUS and GFP expression was observed in 3-week-old leaf explants when wounded with a metal screen compared to a needle and sonication (Figure 6C,D).

Based on the observation 68.11% of leaf explants wounded using a screen were expressing GUS while 74.11% of leaf explants expressed the GFP gene. The presence of phenolic compounds, such as acetosyringone, after wounding, helps to activate the *vir*-inducing components, increasing the transformation efficiency (Figure 7 and Figure 8). However, excess wounding might affect tissue recovery. This may cause poor regeneration of the explant, and decrease the efficiency of obtaining a stable transgenic plant. Previous studies have stated that different wounding methods, such as sonication, might be needed for different types of explants such as embryos, immature cotyledons, apical meristems, calli, and hypocotyls [31].

Acetosyringone has been shown to enhance the transient expression of GUS in different species due to the activation of the *viral* gene [32,33]. The current results convincingly illustrate that the addition of acetosyringone improved the transformation efficiency in leaf explants compared to the control. The results of one study also suggest that acetosyringone augmentation in the co-cultivation medium can considerably increase the transformation efficiency and is necessary for the successful transformation of chickpeas [34]. This study’s results agree with others who demonstrated the successful application of acetosyringone at a concentration of 100 μM in enhancing the transformation efficiency in cowpeas [35] and cucumbers [36].

As hemp is considered recalcitrant to transformation, we also tested the effect of acetosyringone (AS, phenolic inducer) on transient expression efficiency. Several studies have reported that phenolic inducers can enhance the ability of *Agrobacterium* to infect and transform host plants. In genetic transformation, acetosyringone is necessary for the activation of the *Agrobacterium* virulence machinery. It induces virulence genes and facilitates the transfer of T-DNA from a plasmid into the plant cell and its integration into the host genome. The effect of six different concentrations of acetosyringone (0, 50, 11, 150, 200, and 250 µM) was investigated for co-cultivating leaf explants for transformation by subjecting 3-week-old leaves to a co-cultivation lasting 3 days. It was observed that acetosyringone at 150 µM resulted in the highest percentage of explants expressing both GUS and GFP (78.3% and 71.60%, respectively), while the 200 µM concentration resulted in the highest number of GUS and GFP spots per explant (152.78 and 154.56, respectively). The transient expression efficiency decreased drastically for both GUS (38.10%) and GFP (37.30%) at 250 µM (Figure 9 and Figure 10).

### 2.4. Effect of Immersion Period on Transient Expression Efficiency

The immersion period of the explants in the *Agrobacterium* suspension similarly affects the transformation efficiency. A prolonged immersion time might lead to tissue necrosis, browning, and bacterial overgrowth. At the same time, a reduced time may be insufficient for T-DNA transfer. In this study, six different immersion periods (0, 10, 20, 30, 40, 50, and 60 min) were tested and an immersion period of 30 min in the *Agrobacterium* suspension resulted in the highest percentage expression for GUS (79.0%) and GFP (60.70%) with an average of 119.2 GFP- and 132.50 GUS-expressing events (Figure 11 and Figure 12). The lowest efficiency of gene transfer was observed after a 10-min immersion period, possibly due to insufficient time for *Agrobacterium* to infect a large number of cells. The gradual reduction in transient expression efficiency over time can be attributed to the browning of the leaf and or tissue death. Similar to optical density, immersion time may is also dependent on the type of tissue or explant. Young and small tissues might require less immersion time than tissues such as nodal sections.

### 2.5. Effect of Calcium Chloride and Silver Nitrate

Calcium is a macronutrient that is often present in plant media and cell walls and acts as a signaling molecule for changes in cell composition. In this study, five different concentrations of calcium chloride (0.2, 0.5, 0.75, 1, and 1.5 µM) were tested and the highest percentages of GUS- and GFP-expressing explants were observed at a concentration of 0.5 µM. At 0.5 µM, 76.5% of explants were expressing GUS and GFP with approximately 133 spots per explant (Figure 13 and Figure 14). As the concentration of calcium chloride increased, the efficiency of the transient transformation decreased. It has been reported that exogenous calcium in the media can suppress the expression of virulence genes in bacteria [37]. The high amount of exogenous calcium might also be responsible for the fortification of damaged cell walls, thereby limiting *Agrobacterium* infection.

Similarly, silver nitrate has been shown to affect gene transfer efficiencies in *Agrobacterium*-mediated transformation. In this study, five different concentrations of silver nitrate were tested and a concentration of 60 µM resulted in the highest transformation efficiency. The highest expression of GUS was 79.44% of explants and for GFP, 79.75% of explants showed gene expression. As the concentration of silver nitrate was increased (120 µM), the gene transfer efficiency decreased for both GUS and GFP (43.9% and 43.25%, respectively) (Figure 15 and Figure 16). Silver nitrate is a known antioxidant and ethylene inhibitor that affects cell division. The phenomenon of the gradual increase in transformation efficiency at low concentrations and decrease at high concentrations has been reported in several plants like apple, banana, *Phalaenopsis violacea*, and *Primula vulgaris* [38,39,40,41]. A lower transformation efficiency at higher concentrations can be attributed to the antimicrobial properties of silver ions, as it has been previously used to prevent *Agrobacterium* overgrowth and to prevent the browning of explants [41]. 

## 3. Discussion

Tissue culture and transformation are the most widely used techniques in plant functional genomics studies. Although these techniques have been implemented in several plant species over the decades, there are still recalcitrant crops like *C. sativa*, which are less amenable to these methods. There have been a handful of studies aimed at optimizing both stable and transient gene expression in *C. sativa*. One of the key factors for developing any expression system in recalcitrant crops is to identify all factors affecting transformation efficiency. To achieve this, an agroinfiltration system for transient gene expression is ideal, as a wide range of parameters can be tested in relatively less time. A recent transient expression study in *C. sativa* has demonstrated that several hemp organs can express GUS and GFP genes. The addition of Silwett L-77 and the use of sonication followed by vacuum infiltration significantly improved GUS expression in several tissues and organs [2]. Similarly, in this study, the transient expression protocol was successful in several tissue types and resulted in high GFP and GUS expression.

The type, source, and physiological condition of the tissue play a critical role in *Agrobacterium*-mediated gene transformation. Different explants, such as shoot tips and hypocotyls, from in vitro-grown hemp seedlings have been used. Wahby et al. [23] used different parts of 5-day-old in vitro-grown seedlings such as hypocotyls, cotyledons, cotyledonary nodes, and primary leaves for gene transformation, and their report states that the best gene transformation results were obtained from hypocotyl segments. Sorokin et al. [42,43] also used cotyledons and true leaves of 4-day-old in vitro-grown hemp seedlings for gene transformation. Deguchi et al. [2] reported a successful gene transformation using male and female flowers, stems, leaves, and root tissues derived from 2-month-old in vitro-grown hemp seedlings. In the present study, we observed that 3-week-old fully expanded wounded leaf explants had 76.22% GUS expression with an average of 211.0 blue spots per explant compared to 76.13% GFP expression and an average of 201.67 green fluorescence spots. Feeney and Punja [22] used callus cells derived from stem and leaf segments of *Cannabis* for *Agrobacterium*-mediated gene transformation. In the present study, three-week-old leaf-derived calli showed 81.56% GUS expression with an average of 49.22 blue spots per callus piece and 45.22% of callus pieces showed GFP expression with an average of 64.67 green fluorescence events.

It is well known that the optimum density of the *Agrobacterium* culture directly affects the stable and transient transformation efficiency [35]. Depending on factors such as the *Agrobacterium* strain, viability, tissue type, size, cell wall, and wounding method, different types of explants might require different densities for infecting a maximum number of cells. The co-cultivation period and concentration of the *Agrobacterium* inoculum (optical density (OD)) have a significant impact on successful gene transformation. Feeney and Punja [22] suggested three days of co-cultivation and an OD_600nm_ of 1.6–1.8 for gene transformation of callus cells. In another study, different explants were co-cultured for two days [23]. Sorokin et al. [42] suggested three days of co-cultivation and an OD_600nm_ of 0.6 for gene transformation of different parts of in vitro-grown hemp seedlings. In this study, it was observed that the GUS and GFP expression efficiency increased from OD_600nm_ = 0.2 to 0.4 in leaf tissues after three days of co-cultivation. The efficiency gradually decreased as the optical density increased to OD_600nm_ = 1.0. This decrease in efficiency can be attributed to possible tissue damage or cell death due to high bacterial infection, resulting in a lack of gene expression. However, it also needs to be noted that the higher *Agrobacterium* density might lead to higher transient expression but might make it difficult for tissue recovery or regeneration in the case of stable expression [44,45,46,47].

The immersion period is another factor that affects transient expression based on the tissue type, *Agrobacterium* strain, and wounding method. A sufficient immersion period is required for a maximum number of cells to become infected by *Agrobacterium* and in this study, a 30-min immersion period resulted in the maximum transient expression in leaf tissues. Similar to optical density, the transformation efficiency decreased with an increase in the immersion period. Methods such as the use of surfactant, stirring, or vacuum during immersion are known to aid the access of *Agrobacterium* to the plant cell, thereby increasing transformation efficiency.

The tissue wounds allow *Agrobacterium* to be effective in the plant cell [13]. In the case of recalcitrant plants, identifying the wounding method that provides several entry points for *Agrobacterium* without killing or affecting the regeneration potential is necessary. In this study, needle wounding, gentle wounding with a 60 µM screen, and sonication were tested and it was found that leaf tissues wounded with the screen had higher relative GUS and GFP expression compared to the other techniques. The wounding of plant tissues also releases phenolic compounds that attract *Agrobacterium* and induce virulence genes. Various phenolic compounds are produced at the wounding site that can activate the expression of the *vir* gene in *Agrobacterium* and lead to the recognition of *Agrobacterium* by plant cells and subsequent infection [30]. Hence, the addition of phenolic compounds, such as acetosyringone, to the co-cultivation media has been shown to increase transformation efficiency. In the current study, six different acetosyringone concentrations were tested and it was found that co-cultivation media supplemented with 150 µM acetosyringone resulted in the highest percentage of tissue expressing GUS and GFP. The transformation efficiency decreased with an increase in acetosyringone concentration, which might be attributed to phenolic toxicity to plant tissues in higher concentrations.

Silver nitrate is a known antioxidant and inhibits ethylene synthesis. During *Agrobacterium*-mediated transformation, ethylene production is increased as well as the accumulation of ROS due to tissue wounding. Both factors are known to cause tissue necrosis that hinders *Agrobacterium* infection and reduces efficiency. To inhibit ethylene and scavenge ROS, supplementation with five different concentrations of silver nitrate was tested. A 60 µM concentration of silver nitrate had a significant positive effect on the transient expression of both GUS and GFP. These results are consistent with studies conducted in banana and *Phalaenopsis violacea* [13,40]. Similarly, we also tested different concentrations of calcium chloride. Calcium is an important macronutrient and plays an important role in cell wall repair. A reduced concentration of calcium leads to the deterioration of the cell wall. The current study found supplementation with 0.5 µM calcium chloride improved transient transformation efficiency.

## 4. Materials and Methods

### 4.1. Seed Sterilization and Establishment of In Vitro Cultures

The *C. sativa* ‘Joey’ variety is a dual-purpose monecious variety grown for its grain and fiber. It has gone through several breeding generations with some heterogeneity due to its monecious characteristics. The flowering period is typically 55–70 days with the CBD content ranging from 1 to 2%. This variety was developed in Canada and was purchased from Parkland Industrial Hemp Growers Coop, Ltd. (3-126 Main St. N., Dauphin, MB R7N 1C2, Canada). The hemp strain ‘Joey’ was grown following the approved guidelines for industrial hemp provided by the Pennsylvania Department of Agriculture Bureau of Plant Industry under the regulated permits IH-16-P-2017 and IH-17-P-2017. The cultivar ‘Joey’ is an established line that was produced by several rounds of breeding. There is some phenotypic heterogeneity expected in the seedlings derived from the clones of ‘Joey’ in their response to the various parameters that were evaluated in this study. The seeds were harvested from mature plants under controlled conditions in the greenhouse at the University of Pennsylvania campus facilities (Middletown, PA, USA) [2,7,25]. The research experiments were conducted at Fort Valley State University and were approved by the Georgia Department of Agriculture (GDA), and followed the State of Georgia by House Bill 213 Georgia Hemp Farming Act, SB 195 and SB 324, and the University System of Georgia’s institutional policies and procedures. The mature seeds were rinsed with water for 20 min and surface-sterilized in an aqueous 30% Clorox (Clorox pro, 8.25% sodium hypochlorite) solution containing 3 drops of Tween-20 for 30 min on a shaker, and then rinsed 3 times, 5 min each, with sterile water. The sterilized seeds were soaked in sterile 1% hydrogen peroxide overnight to soften the seed coat and increase the rate of seed germination.

After soaking, the outer dark brown hard seed coats and inner dark green membranous coats were removed under sterile conditions with the aid of pointed tweezers and a surgical knife. Care was taken not to cause any injury to the naked seeds (including cotyledons, radicles, and embryos). The sterilized seeds were then germinated on a half-strength Murashige and Skoog’s (MS) basal medium [48] with vitamins containing 3% (*w*/*v*) sucrose, 10 mg L^−1^ thiamine, 100 mg L^−1^ myoinositol (pH 5.8), and solidified with 0.3% (*w*/*v*) Gelrite^®^. All the cultures were incubated at 26 ± 2 °C under a 16/8 h (light/dark) photoperiod with light provided by cool-white, fluorescent lamps at an intensity of 31–35 µmol m^−2^ s^−1^. The explants from one- to three-week-old seedlings were used for genetic transformation.

### 4.2. The Effect of Explants and Developmental Stages

For the transformation experiment, six different types of explants from three-week-old seedlings were used, such as leaves, cotyledons, hypocotyls, root segments, nodal parts, and leaf-derived embryogenic calli. Leaf explants at three developmental stages (1-, 2-, and 3-week-old fully expanded leaf from seedling) were used to investigate the effect of the explant and developmental stages on the transient transformation efficiency. In a separate experiment, the fully expanded 3-week-old leaves were cut into sections (1 cm each) and used as explants for embryogenic callus induction on Gamborg B5 basal salt medium [49] supplemented with 3.0% maltose, 5.1 mM calcium chloride, 5.5 mM glutamine, 32.5 µM glutathione, 95.1 µM serine, 7.4 µM adenine, 4.5 µM 2,4-D, 0.9 µM kinetin, and 0.3% Gelrite [50]. The pH of the medium was adjusted to 5.80 with 1 N NaOH or 1.0 N HCl and then autoclaved at 100 kPa (121 °C) for 20 min. After 4 weeks of culturing in the dark at 26 ± 2 °C, the effect of each treatment on callus induction was surveyed. All cultures were incubated at 26 ± 2 °C under a 16/8 h (light/dark) photoperiod with light provided by cool-white, fluorescent lamps at an intensity of 31 to 35 mol m^−2^ s^−1^.

### 4.3. Preparation of Agrobacterium tumefaciens for Transformation

*A. tumefaciens* strain GV3101 transformed with plasmid pCambia 1304 expressing both *β-glucuronidase* (GUS) and Green Fluorescent Protein (*GFP*) genes were used for transformation. The GFP-GUS gene fusion was under the control of a *Cauliflower mosaic virus* (CaMV) 35S promoter and NOS terminator. The T-DNA region of the plasmid also contained the hygromycin B phosphotransferase gene driven by the CaMV 35S promoter. The pCambia 1304 plasmid has been routinely used to demonstrate or establish transient expression in several plant species such as *Arabidopsis* [51], several medicinal plants [52], rice [53], tobacco plant [54], and *Dendrobium Sonia* [55], and tobacco leaves [56]. The vector pCambia 1300 was used as a control. A single *Agrobacterium* colony derived from the stock culture described by Dan et al. [57] was cultured in liquid Luria Bertani (LB) [58] medium containing 50 mg L^−1^ kanamycin and was grown on a shaker at 120 rpm at a temperature of 28 °C for 16 h. The bacterium was pelleted at 5000× *g* for 10 min, washed in antibiotic-free LB medium, re-pelleted, and re-suspended in fresh LB medium twice. The control consisted of *Agrobacterium* which was not subcultured and was used directly for transformation after centrifugation and resuspension in liquid LB medium. Once the preferred OD was achieved, 100 μM acetosyringone was added to the bacterial suspension culture to increase the virulence. Explants of different ages were transferred into the *Agrobacterium* suspension for transformation. The explants were immersed in the *Agrobacterium* cell suspension for 30 min at 26 °C and gently shaken on a rotary shaker at 50 rpm to ensure that the entire explants were fully submerged for bacterial adherence. After different immersion periods (0, 10, 20, 30, 40, or 60 min), the explants were subsequently washed with sterile distilled water containing 200 mg L^−1^ cefotaxime to remove any excess bacteria. The infected explants were blot-dried using sterile Whatman No. 1 filter paper for 5 min and then the explants were transferred to the co-cultivation medium. The explants were then inoculated onto the pre-selection medium (MS containing 0.23 mg L^−1^ BA, 50 mg L^−1^ cefotaxime, and 100 mg L^−1^ carbenicillin) and incubated at 26 ± 2 °C for 3 days in the dark for co-cultivation. After 3 days, the explants were washed 4 to 5 times with sterile distilled water, immersed in a 200 mg L^−1^ carbenicillin solution for 5 to 10 min, and blotted dry using sterilized filter paper.

### 4.4. Evaluation of Factors Affecting Transient Transformation

To assess the factors affecting transformation efficiency, different treatments were analyzed. All the experiments were conducted with at least 3 replicates with each containing 25 to 30 samples for optimizing the individual parameter. The parameters included explants derived from 3-week-old seedlings such as a leaf, cotyledon, hypocotyl, root, nodal part, and leaf-derived 2- and 3-week-old embryogenic callus, age of explant (1- to 3-week-old fully expanded leaf), immersion period (0, 10, 20, 30, 40, and 60 min), acetosyringone concentration (0, 50, 11, 150, 200, and 250 µM) during co-cultivation, bacterial density at 600 nm (0.2, 0.4, 0.6, 0.8 and 1.0), calcium chloride concentration (0, 0.2, 0.5, 0.75, 1.0 and 1.5 µM), and silver nitrate concentration (0, 30, 60, 90, and 120 µM).

### 4.5. Influence of Wounding Treatment on Transient Gene Expression

To test the influence of different wounding treatments, similar fully expanded 3-week-old leaf explants were gently stabbed 4 to 5 times using a sterile hypodermic needle (23 gauge), and/or a 60 µm stainless steel screen was pressed over the surface of the leaf. For the sonication treatment, the explants were immersed in 30 mL of *Agrobacterium* suspension and sonicated for 30 s twice with a 15 s interval between each treatment [Fisherbrand Model 120 Sonic Dismenbrator]. After wounding, the explants were transferred into a fresh *Agrobacterium* suspension and incubated at 30 °C for a further 30 min for the infection time with occasional gentle agitation. After co-cultivation, the explants were washed 3 to 4 times with liquid MS medium and blotted dry on sterile filter paper. Then, the explants were placed on culture media for three days in the dark at 25 ± 2 °C.

As a control, the explants were directly placed on the co-cultivation medium without being immersed in the *Agrobacterium* suspension. At the end of the co-cultivation period, the histochemical GUS assay and fluorescence microscopy were performed. A total of 25 to 30 explants were used for each treatment, and the experiments were repeated 3 to 4 times. All the parameters were evaluated based on observations of the average percentage of both GUS- and GFP-expressing explants and the number of GUS- and GFP-expressing spots per explant.

### 4.6. Histochemical GUS Assay and Fluorescence Microscopy

The method for the histochemical GUS assay was adopted from [59] with some modifications. The explants were incubated overnight at 37 ± 2 °C in a buffer containing 0.5 mg mL^−1^ 5-bromo-4-chloro-3-indolyl-β-D-glucuronic acid (X-Gluc), 0.1 M sodium phosphate buffer (pH 7.0), 0.1 M potassium ferricyanide, 0.1 M potassium ferrocyanide, and 0.1% (*v*/*v*) Triton X-100. The X-Glue solution was removed, and the explants were washed with double distilled water followed by a wash with 70% ethanol. To remove the chlorophyll and other pigments, a mixture of acetone and methanol (1:3) was added and explants were incubated at 4 °C for 1 h before being visually inspected. After the tissue became transparent, the explants were washed with distilled water twice and stored in 50% glycerol [60,61]. Green fluorescent protein expression in leaves, hypocotyl segments, calli, and developing embryos was visualized using an Olympus SZX12 Stereo Fluorescence Microscope equipped with an HBO 100 W mercury bulb light mounted with a long pass GFP filter and DP74 Cooled Color camera (Olympus America Inc., Melville, NY, USA). The GFP filter had excitation wavelengths of 460–480 nm and emission wavelengths of 495–540 nm, and the GFPA filter for the separation of GFP and blue-excited fluorophores had excitation wavelengths of 460–490 nm and emission wavelengths of 510–550 nm. Photographs were taken using an Olympus SZX12 Automatic Exposure Photomicrographic System.

### 4.7. Statistical Analysis

To optimize the individual parameter, each experiment contained at least 25 to 30 samples, and each experiment was replicated three times (*n* = 75). The experimental data were statistically analyzed using analysis of variance (ANOVA). The treatment means were separated by using the Tukey–Kramer honestly significance difference (HSD) test at *p* ≤ 0.05 [62].

## 5. Conclusions

In this study, a protocol for the optimization of several factors of transient expression in industrial hemp (*C. sativa*) using both GFP and GUS as marker genes was established using 3-week-old seedlings. Different parts of the seedling were used, such as the cotyledon, hypocotyl, root segment, nodal part, and 2- to 3-week-old leaf-derived calli. The improved parameters from this study can be used to improve both the transient and stable expression of genes. They could also enable the implementation of genome editing through CRISPR/Cas systems for *C. sativa* breeding, promoting the development of varieties with enhanced agronomic and medicinal properties with industrial and pharmacological (metabolic engineering) utility.

## Figures and Tables

**Figure 1 plants-13-00664-f001:**
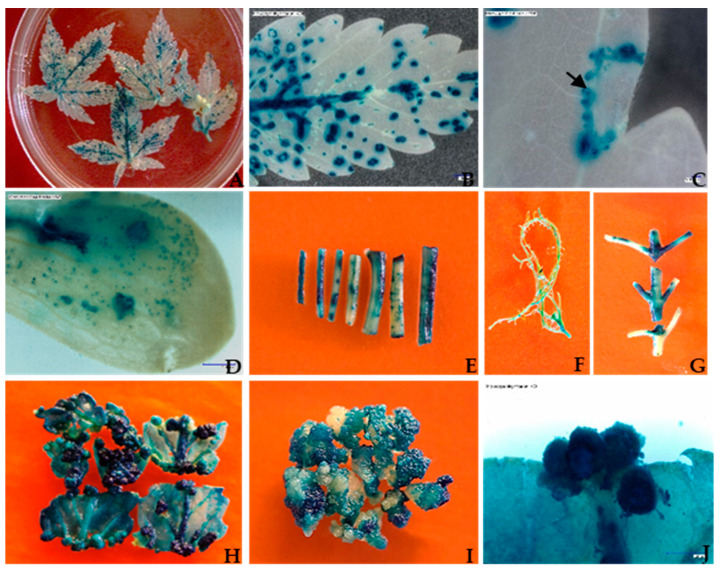
Histochemical GUS assays were performed on different transformed explants of *Cannabis sativa* L. Explants were transformed with the binary vector pCAMBIA1304 via *Agrobacterium tumefaciens* (strain GV3101). After 3 days of co-cultivation, histochemical staining for the GUS activity was performed to measure transient gene expression. The dark-blue-stained tissue indicates the presence of GUS activity. The experiments were repeated three times. (**A**) Three-week-old fully expanded wounded leaves using screen; (**B**) high-magnification (10 × 20; scale bar = 1.0 mm) image showing dark blue staining in three-week-old leaf tissues; (**C**) *β-glucuronidase* (GUS) gene expression in a single cell and multiple cells in leaf tissues. The arrow indicates the expression of the GUS gene in single to multiple cells in leaf tissues. (magnification = 10 × 100; scale bar = 100 µm); (**D**) seven-day-old seedling cotyledon with transient *β-glucuronidase* expression, indicated by dark blue staining at various levels (magnification = 10 × 5); (**E**) Seven to fourteen-day-old seedling hypocotyl expressing the *β-glucuronidase* gene; (**F**) the GUS gene expression in fourteen -day-old seedling roots; (**G**) expression of the *β-glucuronidase* gene in fourteen -day-old seedling nodal part; (**H**) two-week-old leaf-derived embryogenic callus pieces expressing the *β-glucuronidase* gene; (**I**) three-week-old leaf-derived callus culture with evident GUS-carrying globular embryo development; (**J**) three-week-old leaf-derived globular-shaped embryo under high magnification showing *β-glucuronidase* gene expression.

**Figure 2 plants-13-00664-f002:**
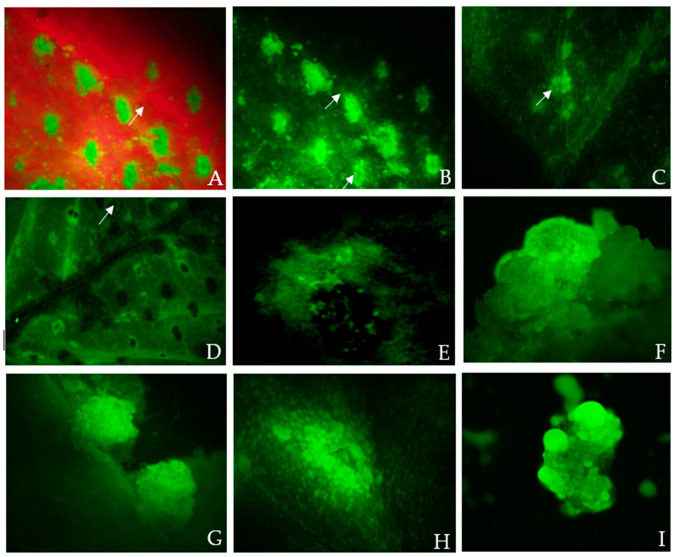
GFP protein fluorescence in *Cannabis sativa* L. wounded fully expanded 3-week-old leaf explants and leaf-derived embryogenic calli co-cultivated for 3 days with *Agrobacterium* suspension at final OD_600nm_ of 0.6 and washed 5× times with Murashige and Skoog washing medium before capturing the images using an Olympus SZX12 Stereo Fluorescence Microscope (Olympus America Inc., Melville, NY, USA). White arrows indicate GFP gene expression. The images were obtained using the fluorescence microscope mounted with a long-pass GFP filter and a DP72 camera. The GFP filter had excitation wavelengths of 460 to 480 nm and emission wavelengths of 495 to 540 nm, and the GFPA filter for the separation of GFP and blue-excited fluorophores had excitation wavelengths of 460 to 490 nm and emission wavelengths of 510 to 550 nm; these were used to identify the leaf and embryogenic callus tissues expressing GFP. (**A**,**B**) show the corresponding fluorescence image of 3-week-old fully expanded wounded leaves (cluster of cells) exhibiting GFP fluorescence and chloroplast autofluorescence, captured using a stereo fluorescence microscope under a GFP filter. The leaf tissues displayed green fluorescence that was partially masked by red fluorescence from chlorophyll (scale bar = 500 µm). (**C**) GFP protein fluorescence within the clusters of multiple cells in the leaf tissue (scale bar = 300 µm, GFPA filter); (**D**) strong expression of the GFP gene in primary vascular structures 3 days after co-cultivation (arrow); (**E**) strong expression of the GFP gene in the multiple isolated cells; (**F**,**G**) three-week-old leaf-derived callus pieces grown in a non-selection medium expressing GFP (globular embryos) (scale bar = 500 µm); (**H**) transformed globular-shaped embryogenic calli grown in a non-selection medium showing GFP expression (scale bar = 500 µm); (**I**) transformed nodular calli grown in the selection medium (10 mg/L hygromycin) showing GFP expression on day 14. (**D**,**E**,**I**) Magnification = 10 × 40.

**Figure 3 plants-13-00664-f003:**
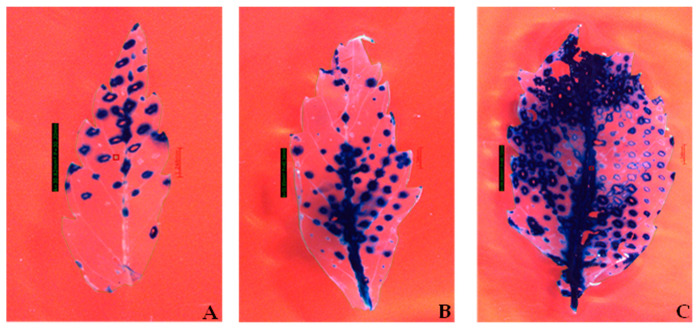
Histochemical GUS assays were performed on transformed *Cannabis sativa* L. leaf tissues at different stages of development. One- to three-week-old leaves were transformed with the binary vector pCAMBIA1304 via *Agrobacterium tumefaciens* (strain GV3101). After 3 days of co-cultivation, histochemical staining for the GUS activity was performed. The dark blue staining indicates transient GUS gene expression. The experiments were repeated three times. (**A**) Expression of the *β-glucuronidase* gene in 1-week-old leaves with a surface area of 30.20 mm^2^. (**B**) Expression of the *β-glucuronidase* gene in 2-week-old leaves with a surface area of 55.85 mm^2^. (**C**) Expression of the *β-glucuronidase* gene in 3-week-old leaves with a surface area of 99.04 mm^2^. Pictures were taken using a Leica EMSPIRA/3 Stereo Zoom Microscope (Leica Microsystems Inc., Deerfield, IL, USA) at a magnification of 0.743× (scale bar = 1.248 mm). The calculated surface area of the 1- to 3-week-old explants is indicated on the left side of (**A**–**C**).

**Figure 4 plants-13-00664-f004:**
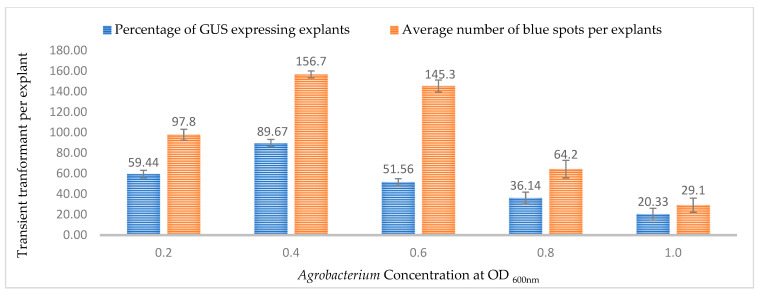
Effect of bacterial density at OD_600nm_ = 0.6 on transformation efficiency based on transient *β-glucuronidase* (GUS) expression. Infection frequency was calculated as the percentage of GUS-positive explants from 3-week-old leaves out of the total number of explants examined. The number of GUS foci per explant is the average number of GUS-positive foci in at least three independent explants. The data are represented as means ± standard error (SE). To optimize the individual parameter, each experiment contained 25 to 30 samples and each experiment had 3 replications (*n* = 75).

**Figure 5 plants-13-00664-f005:**
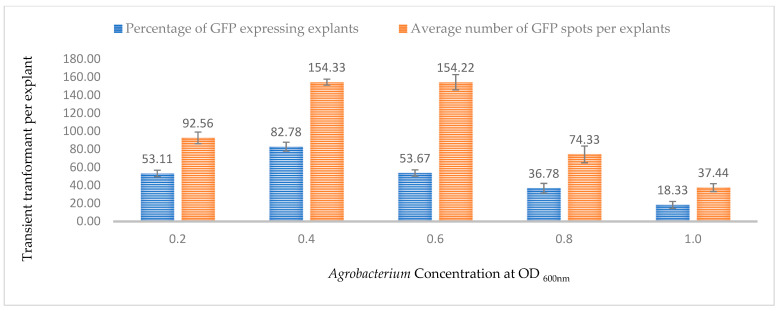
Effect of bacterial density at OD_600nm_ = 0.6 on transformation efficiency based on GFP expression. Infection frequency was calculated as the percentage of GFP-positive explants from 3-week-old leaves out of the total number of explants examined. The number of GFP foci per explant is the average number of GFP-positive foci in at least three independent explants. The data are represented as means ± standard error (SE). To optimize the individual parameter, each experiment contained 25 to 30 samples and each experiment had 3 replications (*n* = 75).

**Figure 6 plants-13-00664-f006:**
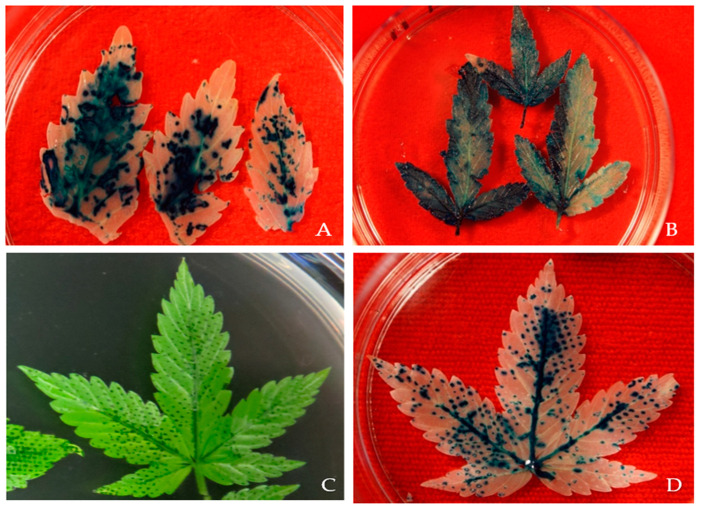
Effect of wounding on GUS gene expression. Histochemical GUS assays were performed on transformed explants of *Cannabis sativa* L. Explants were transformed with the binary vector pCAMBIA1304 (1) via *Agrobacterium tumefaciens*. Explants were co-cultivated for 3 days with *Agrobacterium*, followed by histochemical staining for GUS activity. The dark blue staining indicates GUS activity in the various explants. The experiments were repeated three times with similar results. Three-week-old fully expanded and physically wounded by (**A**) using a hypodermic 23-gauge needle; (**B**) sonicated for 30 s twice with a 15 s interval; (**C**) physically pressing with 60 μM screen; (**D**) dark blue GUS staining in 3-week-old leaf tissues.

**Figure 7 plants-13-00664-f007:**
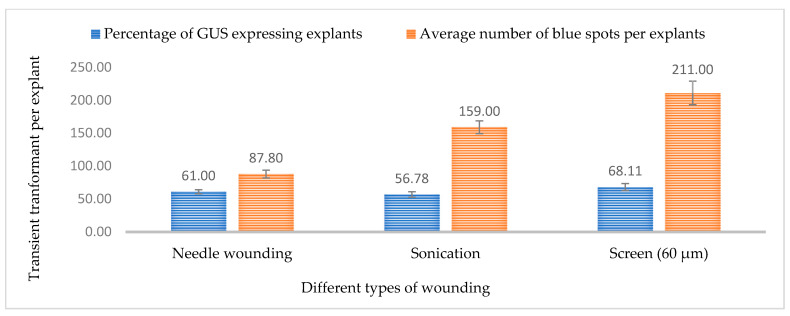
Effect of different wounding methods on transformation efficiency based on transient GUS expression. Infection frequency was calculated as the percentage of GUS-positive explants from 3-week-old leaves out of the total number of explants examined. The number of GUS foci per explant is the average number of GUS-positive foci in at least three independent explants. The data are represented as means ± standard error (SE). To optimize the individual parameter, each experiment contained 25 to 30 samples and each experiment had 3 replications (*n* = 75).

**Figure 8 plants-13-00664-f008:**
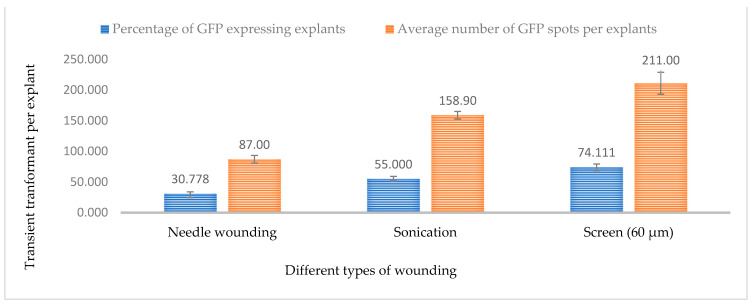
Effect of different wounding methods on transformation efficiency based on GFP expression. Infection frequency was calculated as the percentage of GFP-positive explants from 3-week-old leaves out of the total number of explants examined. The number of GFP foci per explant is the average number of GFP-positive foci in at least three independent explants. The data are represented as means ± standard error (SE). To optimize the individual parameter, each experiment contained 25 to 30 samples and each experiment had 3 replications (*n* = 75).

**Figure 9 plants-13-00664-f009:**
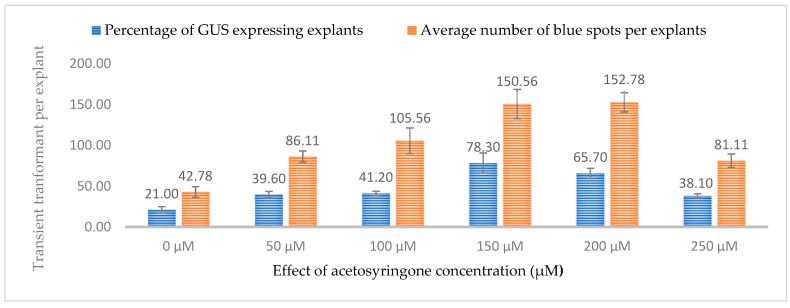
Influence of acetosyringone concentrations (µM) on transient GUS expression. Infection frequency was calculated as the percentage of GUS-positive explants from 3-week-old leaves out of the total number of explants examined. The number of GUS foci per explant is the average number of GUS-positive foci in at least three independent explants. The data are represented as means ± standard error (SE). To optimize the individual parameter, each experiment contained 25 to 30 samples and each experiment had 3 replications (*n* = 75).

**Figure 10 plants-13-00664-f010:**
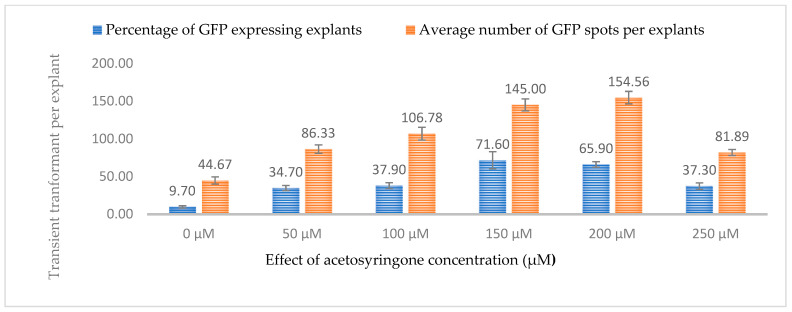
Influence of acetosyringone concentrations (µM) on transient GFP expression. Infection frequency was calculated as the percentage of GFP-positive explants from 3-week-old leaves out of the total number of explants examined. The number of GFP foci per explant is the average number of GFP-positive foci in at least three independent explants. The data are represented as means ± standard error (SE). To optimize the individual parameter, each experiment contained 25 to 30 samples and each experiment had 3 replications (*n* = 75).

**Figure 11 plants-13-00664-f011:**
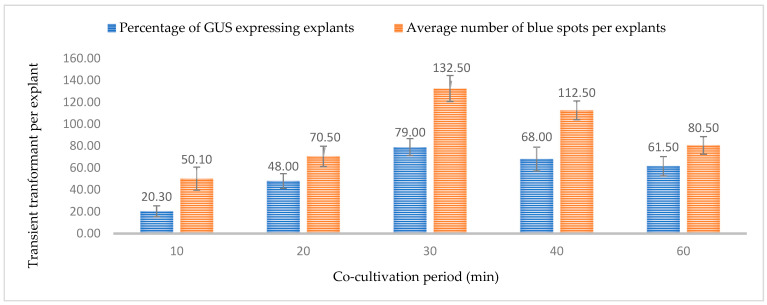
Effect of immersion period on transformation efficiency based on transient *β-glucuronidase* (GUS) expression. Infection frequency was calculated as the percentage of GUS-positive explants from 3-week-old leaves out of the total number of explants examined. The number of GUS foci per explant is the average number of GUS-positive foci in at least three independent explants. The data are represented as means ± standard error (SE). To optimize the individual parameter, each experiment contained 25 to 30 samples and each experiment had 3 replications (*n* = 75).

**Figure 12 plants-13-00664-f012:**
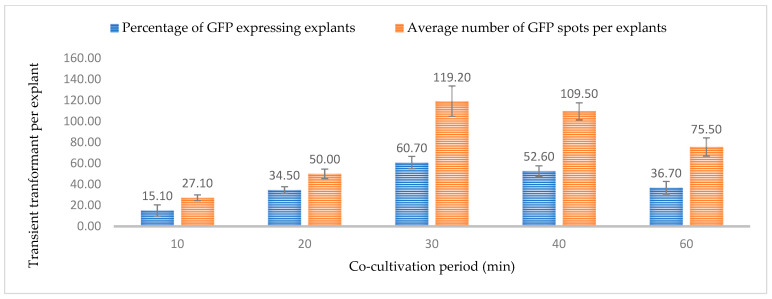
Effect of immersion period on transformation efficiency based on Green Florescent Protein (GFP) expression. Infection frequency was calculated as the percentage of GFP-positive explants from 3-week-old leaves out of the total number of explants examined. The number of GFP foci per explant is the average number of GFP-positive foci in at least three independent explants. The data are represented as means ± standard error (SE). To optimize the individual parameter, each experiment contained 25 to 30 samples and each experiment had 3 replications (*n* = 75).

**Figure 13 plants-13-00664-f013:**
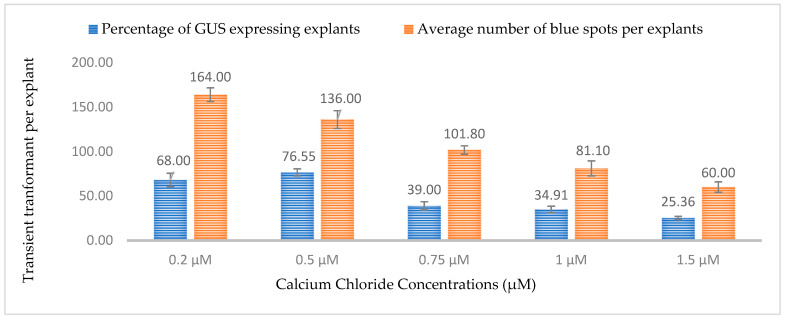
Effect of different calcium chloride concentrations on transformation efficiency based on transient *β-glucuronidase* (GUS) expression. Infection frequency was calculated as the percentage of GUS-positive explants from 3-week-old leaves out of the total number of explants examined. The number of GUS foci per explant is the average number of GUS-positive foci in at least three independent explants. The data are represented as means ± standard error (SE). To optimize the individual parameter, each experiment contained 25 to 30 samples and each experiment had 3 replications (*n* = 75).

**Figure 14 plants-13-00664-f014:**
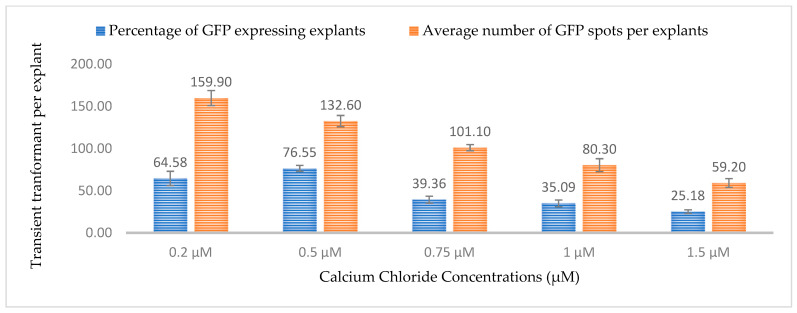
Effect of different calcium chloride concentrations on transformation efficiency based on transient GFP expression. Infection frequency was calculated as the percentage of GFP-positive explants from 3-week-old leaves out of the total number of explants examined. The number of GFP foci per explant is the average number of GFP-positive foci in at least three independent explants. The data are represented as means ± standard error (SE). To optimize the individual parameter, each experiment contained 25 to 30 samples and each experiment had 3 replications (*n* = 75).

**Figure 15 plants-13-00664-f015:**
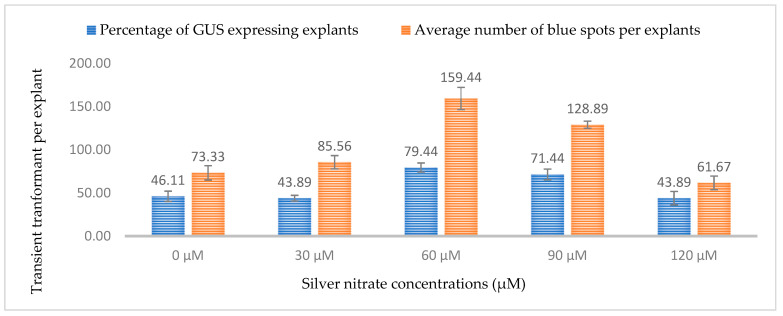
Effect of different silver nitrate concentrations on transformation efficiency based on transient *β-glucuronidase* (GUS) expression. Infection frequency was calculated as the percentage of GUS-positive explants from 3-week-old leaves out of the total number of explants examined. The number of GUS foci per explant is the average number of GUS-positive foci in at least three independent explants. The data are represented as means ± standard error (SE). To optimize the individual parameter, each experiment contained 25 to 30 samples and each experiment had 3 replications (*n* = 75).

**Figure 16 plants-13-00664-f016:**
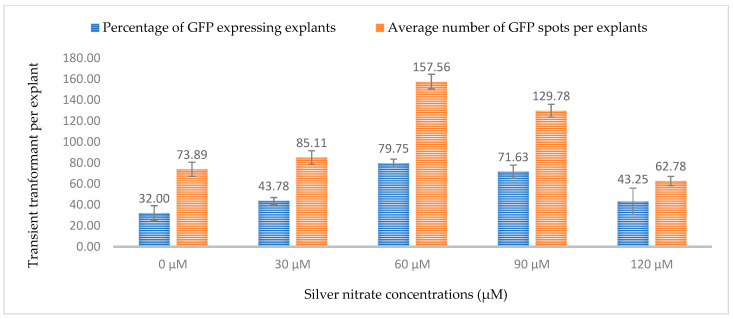
Effect of different silver nitrate concentrations on transformation efficiency based on Green Florescent Protein (GFP) expression. Infection frequency was calculated as the percentage of GFP-positive explants from 3-week-old leaves out of the total number of explants examined. The number of GFP foci per explant is the average number of GFP-positive foci in at least three independent explants. The data are represented as means ± standard error (SE). To optimize the individual parameter, each experiment contained 25 to 30 samples and each experiment had 3 replications (*n* = 75).

**Table 1 plants-13-00664-t001:** Transformation efficiency in different types of explants based on transient *β-glucuronidase* (GUS) expression.

Type of Explant	* The Average Percentage of GUS-Expressing Explants	** Average No. of Blue Spots per Explant
1-week-old leaves 2-week-old leaves 3-week-old leaves	35.0 ± 9.91 54.4 ± 3.90 76.22 ± 6.08	87.8 ± 6.18 159.0 ± 9.17 211.0 ± 5.67
Cotyledon	49.78 ± 6.16	81.67 ± 7.91
Hypocotyl	40.78 ± 6.02	60.0 ± 7.10
Root segment	43.33 ± 7.14	66.0 ± 9.50
Nodal part	53.44 ± 5.29	26.89 ± 8.04
2-week-old leaf-derived callus	54.67 ± 5.41	52.44 ± 15
3-week-old leaf-derived callus	81.56 ± 6.48	49.22 ± 3.93

* Infection frequency was calculated as the percentage of GUS-positive explants from 3-week-old seedlings out of the total number of explants examined. ** The number of GUS foci per explant is the average number of GUS-positive foci in at least three independent explants. The data are represented as the mean ± standard error (SE). To optimize the individual parameter, each experiment contained 25 to 30 samples and each experiment had 3 replications (*n* = 75).

**Table 2 plants-13-00664-t002:** Transformation efficiency in different types of explants based on transient Green Florescent Protein (GFP) expression.

Type of Explant	* The Average Percentage of GFP-Expressing Explants	** Average No. of Green (GFP) Spots per Explant
1-week-old leaves 2-week-old leaves 3-week-old leaves	36.13 ± 4.36 53.50 ± 4.21 76.13 ± 3.91	86.33 ± 7.79 154.22 ± 10.92 201.67 ± 15.21
Cotyledon	48.33 ± 6.06	81.67 ± 7.91
Hypocotyl	34.44 ± 2.74	60.0 ± 7.10
Root segment	41.22 ± 6.10	46.11 ± 5.56
Nodal part	49.11 ± 2.03	29.00 ± 6.02
2-week-old leaf-derived callus	36.56 ± 3.50	52.44 ± 15.0
3-week-old leaf-derived callus	45.22 ± 6.87	64.67 ± 9.07

* Infection frequency was calculated as the percentage of GFP-positive explants from 3-week-old seedlings out of the total number of explants examined. ** The number of GFP foci per explant is the average number of GFP-positive foci in at least three independent explants. The data are represented as means ± standard error (SE). To optimize the individual parameter, each experiment contained 25 to 30 samples and each experiment had 3 replications (*n* = 75).

## Data Availability

All data are available within the manuscript, and further information can be obtained on request from the corresponding authors.

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
