# Peer review of "Evaluation of Parameters Affecting Agrobacterium-Mediated Transient Gene Expression in Industrial Hemp (Cannabis sativa L.)"

_plants, 2024, doi:10.3390/plants13050664_

Round 1

Reviewer 1 Report (Previous Reviewer 3)

Comments and Suggestions for Authors

Here are my objections again:  

2. Especially in case of transient expression, the choice of expression cassette should be a construct which works exclusively in eukaryotic cells, e.g. https://link.springer.com/article/10.1007/BF00260489 The pCAMBIA1301 gus in ref [37]  does contain a cat1 intron. In case of ref [39] the method of gene transfer for the transient assay is particle bombardment, not agrobacteria, and derivatives of the pCAMBIA1304 were used. Agros wer used for the generation of stable transgenic tobacco lines.  

4. Here, I ment, of course, the bar figures 5-6 and 8-17. In those figures only one type of explant is represented, the other explant types are only listed in fig 1 and 2.  

5. It is not very straightforward just to add the surface area of the leaves, it does not add much to the representation and evaluation of the results either.  

9. There is still no answer, if the authors took the samples from single plants and compared those for transient transformation efficiency or they took a mixture of samples from highly heterogenic hemp plant

Author Response

Thank you very much for taking the time to review this manuscript. Please find the detailed responses below and the corresponding revisions/corrections highlighted/in track changes in the re-submitted files. In the PDF format.

Comments 2: Response: Thank you for pointing this out. Therefore, we have revised the manuscript. Please see the lines: 452-454.

The pCAMBIA1304 has been routinely used to demonstrate or establish the transient expression of several plant species such as Arabidopsis [51], several medicinal plants [52], Rice [53], Tobacco [54], Dendrobium Sonia [55], and tobacco leaves [56]. We have also added 6 additional references [51 to 56], Please see lines: 660-673.

51.  Kim, M.J., Baek, K., Park, C.M. Optimization of conditions for transient Agrobacterium-mediated gene expression assays in Arabidopsis. Plant Cell Rep 2009, 28, 1159–1167. [CrossRef] [Google Scholar]

52.  Xia, P.; Hu, W.; Liang, T.; Yang, D.; Liang, Z. An attempt to establish an Agrobacterium-mediated transient expression system in medicinal plants. Protoplasma. 2020, 257, 1497–1505. [CrossRef] [Google Scholar]

53.  Burman, N.; Chandran, D., Khurana, J.P. A rapid and highly efficient method for transient gene expression in rice plants. Frontiers in Plant Science, 2020, 11, 584011. [CrossRef] [Google Scholar]

54.  Sun, J.; Hu, W.; Zhou, R.; Wang, L.; Wang, X.; Wang, Q.; Feng, Z.; Li, Y.; Qiu, D.; He, G.; Yang, G. The Brachypodium distachyon BdWRKY36 gene confers tolerance to drought stress in transgenic tobacco plants. Plant Cell Rep 2015, 34, 23–35. [CrossRef] [Google Scholar]

55.  Pinthong, R.; Sujipuli, K.; Ratanasut, K. Agroinfiltration for transient gene expression in floral tissues of Dendrobium Sonia ‘Earsakul’. Journal of Agricultural Technology, 2014 10(2), 459-465. [CrossRef] [Google Scholar]

56.  Shafaghi, M.; Maktoobian, S.; Rasouli, R.; Howaizi, N.; Ofoghi, H.; Ehsani, P. Transient Expression of Biologically Active Anti-rabies Virus Monoclonal Antibody in Tobacco Leaves. Iran J Biotechnol. 2018 18;16(1):1774. [CrossRef] [Google Scholar]

Comments: 4. Response: Thank you for pointing this out. As in the abstract [lines 27-30; and Table-1: Line 143-147; and Table-2: Line 149-154],

“In the present study, we tested different explants, such as 1 to 3-week-old leaves, cotyledon, hypocotyl, root segment, nodal part, and 2 to 3-week-old leaf-derived callus. We observed that the 3-week-old leaves are the best explant for transient gene expression”.  Once we observed that 3-week-old leaves are the best explant, for further experiments, we selected 3-week-old leaves as an explant for all experiments”.

Please see Figure 5: lines 212-216; Figure 6: lines 236-242; Figure 8: lines 249; Figure 9:  lines 267-272; Figure 10: lines 273-278; Figure 11: lines 290-295; Figure 12: lines 296-301; Figure 13: lines 310-315; Figure 14: lines 316-321; Figure 15: lines 333-338; Figure 16: lines 339-344.

Comments: 5. Response:  We strongly feel that 1-week-old leaves with a surface area of 30.20 mm2, 2-week-old leaves with a surface area of 55.85mm2 and 3-week-old leaves with a surface area of 99.04mm2, clearly show the comparison on “Percentage of GUS expressing on explant” and “Average number of blue spots on explant”. Since 3-week-old leaves were used for further explants, it also helps us to analyze the age, size, and which part of the leaf area/tissue (top, middle, leaf base, or leaf edges) is ideal for transient gene expression as well as for inducing the regeneration via organogenesis and somatic embryogenesis for stable transformation [manuscript under preparation, unpublished research].  

Comments: 9. Response:  As previously reported by several authors, lines 101-104; reference 27; lines 600-602, for optimization seeding tissue, is ideal for regeneration, so we adapted a similar protocol. Also, please see lines 413-414. “The cultivar 'Joey' is an established line produced by several rounds of breeding. There is some phenotypic heterogeneity expected in the seedlings derived from the clones of ‘Joey’ in their response to various parameters that have been evaluated in this study. The seeds were harvested from mature plants under controlled conditions in the greenhouse at the University of Pennsylvania campus facilities. Please lines Reference 2 and 7; lines: 418-421.

Reviewer 2 Report (New Reviewer)

Comments and Suggestions for Authors

Comments on the Quality of English Language

Author Response

REVIEWER 2: Thank you very much for taking the time to review this manuscript. Please find the detailed responses below and the corresponding revisions/ corrections highlighted/in track changes in the re-submitted files in PDF format. 

Comment 1: How does your current development stand out from currently available reports on the Hemp transformation system? Kindly shed light on this.

Response: Thank you for pointing this out. We think that this article/research constitutes the first report to assess the aforementioned factors in C. sativa ‘Joey’ a dual-purpose monecious variety grown for grain and fiber. This variety was developed in Canada, grown, and seed collected in Pennsylvania, USA; now testing in Georgia, USA for the efficient transient expression of two maker genes GUS and GFP together in different types of tissues such as 1–3-week-old leaves, hypocotyl, cotyledon, root segments, nodal part, and 2–3-week-old leaf derived callus in Cannabis sativa L. The data obtained from previous research in combination with this study will lay the foundation to develop replicable and high-efficiency genetic engineering strategies in different hemp cultivars.

Comment 2: Kindly check the MS thoroughly for spelling mistakes.

Response: As suggested, we have the revised manuscript. 

Comment 3: In my opinion, the surface area can significantly compromise the conclusion. The ratio of several spots to a smaller area and a bigger surface area can have equal results. Kindly think about it while concluding.

Response: Thank you for pointing this out. We strongly feel that 1-week-old leaves with a surface area of 30.20 mm2, 2-week-old leaves with a surface area of 55.85mm2 and 3-week-old leaves with a surface area of 99.04mm2, clearly show the comparison on “Percentage of GUS expressing on explant” and “Average number of blue spots on explant”. Since 3-week-old leaves were used for further explants, it also helps us to analyze the age, size, and which part of the leaf area/tissue (top, middle, leaf base, or leaf edges) is ideal for transient gene expression as well as for inducing the regeneration via organogenesis and somatic embryogenesis for stable transformation [manuscript under preparation, unpublished research].  

Comment 4: I found many repeated figures in this MS Kindly fix this issue.

Response:  We have made the changes in the revised manuscript. 

Comment 5: Some GUS results are strangely similar to GFP results. Is it a coincidence or a mistake in preparing graphs? Kindly check.

Response: Thank you for pointing this out. We have done this in the revised manuscript.   

Comment 6: Repeated figures again.

Response: We have made the changes in the revised manuscript.   

Comment 7: I can’t find any phenolic compounds-related data in Figure 7.

Response: Please see lines 230-235; and reference numbers 32,33,34,35,36.  Acetosyringone has been shown to enhance the transient expression of GUS in different species due to the activation of the viral gene [32,33]. The current results convincingly illustrate that the addition of acetosyringone improved the transformation efficiency in leaf explants compared to the control.     

Comment 8: How do your results suggest that Acetosyrinringone improved transformation efficiency in chickpeas? Aren’t you working on hemp?

Response: Please see lines 260-266; and Figures 9 and 10.   

The effect of six different concentrations of acetosyringone (0, 50, 11, 150, 200, and 250 µM) was investigated for co-cultivating leaf explants for transformation was studied by subjecting 3-week-old leaves with a co-cultivation duration of 3 days. It was observed that acetosyringone at 150 µM resulted in the highest percentage of explants expressing both GUS and GFP (78.3% and 71.60%, respectively), while 200 µM concentration resulted in the highest number of GUS and GFP spots per explant (152.78 and 154.56, respectively) including the number of blue spots and GFP expressing events. The transient expression efficiency decreased drastically for both GUS (38.10%) and GFP (37.30%) at 250 µM.

Comment 9: another repeated figure.

Response: We have made the changes in the revised manuscript. 

Comment 10: Kindly fix the repeated figure problem. I give up here.

Response: We have made the changes in the revised manuscript.   

Reviewer 3 Report (Previous Reviewer 1)

Comments and Suggestions for Authors

The paper entitled 'Evaluation of Parameters Affecting Agrobacterium-mediated Genetic Transformation in Industrial Hemp (Cannabis Sativa L.)' shows optimisation of the transformation process using Agrobacterium tumefaciens.

1.     According to methodology, the experiment was conducted on a large number of samples. In line 477, the authors state that the results are semiquantitative.

Therefore, I have a question why ANOVA was performed, what were the reasons for it, and where are any results of statistical analysis other than SE. If the bars show SE at 25 trials and 3 repetitions, what was the SD. And whether the results are statistically significant in any way. Therefore, is there any justification for the selection of parameters for the new protocol?

2.     What do DS2 and DS3 mean in Tables 1 and 2?

3.     Figures 4,5,7,9,10,13,14 are duplicated.

4.     Figure legends 'The number of GFP foci per explant was the average of GFP-positive foci in at least three independent explants with standard deviation. Data represent means ± standard error (SE).’ Why are SD taken into account sometimes and SE at other times? Why at least 3 explants when ‘To optimise the individual parameter, each 347 experiment contains 25 to 30 samples and each experiment with 3- replication (N=75).’ Is there logic? Or, authors write single, isolated formulas and do not think about what they have written.

5.     Figures 7 and 8 show different results, but the data are identical in many places. Is this a coincidence or a mistake?

6.     There are still typos and errors in the text. e.g., line 464 2H2O.

Author Response

Thank you very much for taking the time to review this manuscript. Please find the detailed responses below and the corresponding revisions/corrections highlighted/in track changes in the re-submitted files as PDF attachments.

Comments 1.  Response: Thank you for pointing this out. Please see the line: 212-215. The number of GFP foci per explant was the average of GFP-positive foci in at least three independent explants with standard deviation. Data represent means ± standard error (SE). To optimize the individual parameter each experiment contains 25 to 30 samples and each experiment with 3- replications (N=75).

Comments 2.  Response: This was during the edit and comments by Sarwan Dhir [DS], we have corrected the revised version of the manuscript in Tables 1 and 2.  

Comments 3. Response: This was a technical error at the time of uploading the revised manuscript on the MSPI publisher’s website, I have attached the revised version of the manuscript in PDF format for the accuracy of the figures.  

Comments 4. Response: Please see the lines: 212-215; and all figure and histogram figure legends. To optimize the individual parameter each experiment contains at least 25 to 30 samples, and each experiment was replicated three times (N=75). We have always used the SD and SE in all the experiments and prepared the histogram.

Comments 5. Response: Please see Figure 7: 243-248 Figure: 249-255 and lines 223-225. The data presented are different in wounding with needle, sonication, and screen (65 um). Based on the observation 68.11% of leaf explants wounded using a screen were expressing GUS while 74.11% of leaf explants expressed the GFP gene. However, we observed the number of GFP spots first using the Olympus SZX12 Stereo Fluorescence Microscope equipped with an HBO 100 W mercury bulb light mounted with a long pass GFP’s filters and later tested the same leaves for the histochemical GUS gene spots. The number of GUS and GFP spots were almost the same using the screen (60 um), this could be due to the expression of both genes at the same level.

Comments 6. Response: Please see the lines: 31, 368, 370, 373, and 528. We have corrected the revised version of the manuscript.  

Reviewer 4 Report (New Reviewer)

Comments and Suggestions for Authors

Review report for plants manuscript Jan,12,2024

General comments: -

This manuscript is not suitable for publication as a research article. The research point is scientifically weak and with a close look at the factors under study, it is easily to find out that all those factors were previously studied as a part of some previous researches as example the OD concentration. The manuscript also needs to be modified and submitted in a clean version .I strongly recommend the author to submit it after a major revision as a short communication where it fits more. 

Detailed comments:

The English language and /writing style needs some moderate changes and grammar check.

Abstract:

This section is well written

Keywords:

-The keywords has been chosen very carefully and accurately.

Introduction:

-The introduction doesn’t provide sufficient background and it is missing enough relevant references.

-This section needs to be elongated and enriched with more up to date background about this topic.

Materials and Methods

It is ok and adequate for the author objective.

 Results:

The results are not enough to be published as a research article.

Discussion:

This section is poorly written.

The author is strongly advised to combine the results and discussion in one section for better interpretation and discussion for the presented  results in Figures and Tables. Please rewrite and combine the results for some factors together to fit the short communication format.

References

  •  

This section needs to be UpToDate.

Comments on the Quality of English Language

Needs some moderate modification 

Author Response

Detailed comments: Response: Thank you for pointing this out. We have corrected the revised version of the manuscript. 

Abstract: This section is well written. Response: Please see lines: 27-30. We have revised the abstract.

Keywords: -The keywords have been chosen very carefully and accurately.

Introduction: Response: Please see the lines: 57-100; 104-116.

We have revised the Introduction section in the manuscript by providing the updated information and including an additional 15 references in the revised version of the manuscript.  References 7-29.

Materials and Methods: Results: 

Response: Please see the lines in the revised version of the manuscript. lines: 114-117.  Thank you for pointing this out. We think that this article/research constitutes the first report to assess the aforementioned factors for the efficient transient expression of two maker genes together in different types of tissues such as 1–3-week-old leaves, hypocotyl, cotyledon, root segments, nodal part, and 2 to 3-week-old leaf derived callus in Cannabis sativa L. The data obtained from previous research in combination with this study will lay the foundation to develop replicable and high-efficiency genetic engineering strategies in different hemp cultivars.

Discussion: Response: We have revised the discussion and have updated the information by adding 12 additional references. Please see lines 357-382; 389; 394-398

 References: Response: We have revised the discussion and have updated the information by adding 20 additional references. Reference: 2,13, 22, 23, 25, 30, 35, 42, 43, 44, 45, 46, 47, 51-58           

Round 2

Reviewer 4 Report (New Reviewer)

Comments and Suggestions for Authors

The new version of the manuscript is well improved and now it is suitable for publication 

Comments on the Quality of English Language

The English sis fine just some minor revision is needed 

This manuscript is a resubmission of an earlier submission. The following is a list of the peer review reports and author responses from that submission.

Round 1

Reviewer 1 Report

Comments and Suggestions for Authors

The paper entitled 'Evaluation of Parameters Affecting Agrobacterium-mediated Genetic Transformation in Industrial Hemp (Cannabis Sativa L.)' shows optimization of the transformation process using Agrobacterium tumefaciens. This bacteria species is a natural plant pathogen that besides A. rhizogenes (formally known as Rhizobium rhizogenes) causes tumor formation in plants (crown gall or hairy roots).

Below are my questions and comments.

1.     Most of all, the authors must improve the description of the experiments and some points of presentation of the results.

2.     In Section 4.4 it was stated that there were the following types of explants (initial material for transformation): leaf, cotyledon, hypocotyl, root, nodal part and leaf-derived embryogenic callus). This does not agree with the resulting part where:

-        On page 2 it says "all" and where in the tables are 1, 2, 3 week old leaves listed?

-        The petiole appears in Table 2 as an explant not mentioned in Section 4.4.

-        In Figure 2, the nodular callus is mentioned, which is not listed in the M&M section.

3.     Figure 2. What is a "nodular calli"? A callus may be undifferentiated, embryogenic, or shoot-differentiating. The place where the callus tissue is formed is not the main factor in its description.

4.     There is no statistical analysis in the tables and graphs. There are only error bars or SE values. The text also gives the following: ‘values with different letters are statistically different (P = 0.05), according to Tukey's test’, but there is not a single letter. Are these results really statistically insignificant? The graphs do not indicate whether it is SE or SD.

5.     All graph headings require supplementing the description with statistics, number of repetitions, etc.

6.     In Section 4.4 it is stated that there were at least 4 replicates of 25 to 30 explants (Do the authors mean samples?). In Section 4.7 it is stated that each experiment was replicated 3 times and in 4.4 that at least 4 times. How many individual trials were finally included in the ANOVA?

7.     A uniform way of writing units should be maintained. M, not mol L-1. I understand mg L-1 but moles should be written the same throughout the article.

8.     kg cm-1 is not an SI unit of pressure, please change it.

9.     How was callus tissue assessed as embryogenic? In section 4.2 you should provide the age of the callus (the results the authors show a callus that is 2 or 3 weeks old). I understand from the methodology that callus was not passaged and conducted as a culture.

10.  I also have a question about whether it was necessary to remove the seed coat. Doesn't hemp germinate without it? The entire process could be limited to sterilisation and placement on a medium after briefly drying on tissue paper.

11.  Isn't the USA required permission to carry out genetic modifications? If so, please provide the permission number.

12.  In the text are some typos and errors in the text. For example, the second part of the species name should be written with a lowercase letter in Latin. Also, superscripts in units or subscripts in chemical formulas need to be checked.

Author Response

Response to the Comments:  

1) Most of all, the authors must improve the description of the experiments and some points of presentation of the results.

Response: Please see: Section Material and Methods: 4.1-4.7 and Result 2. Lines 377-463 for detailed description.  

2) In Section 4.4 it was stated that there were the following types of explants (initial material for transformation): leaf, cotyledon, hypocotyl, root, nodal part, and leaf-derived embryogenic callus). This does not agree with the resulting part where: On page 2 it says "all" and where in the tables are 1, 2, and 3-week-old leaves listed?

Response: Please see updated Tables 1 and 2 and Sections 2.1-2.2: lines 103, Figure 2: lines 145-146.

3) The petiole appears in Table 2 as an explant not mentioned in Section 4.4.

Response: Please see revised Tables 2 and sections 4.2 and lines 393-396.

4) In Figure 2, the nodular callus is mentioned, which is not listed in the M&M section.

Response: Please see: Table 1 and 2, Figure 2 lines 145-146; Material & Method 4.2: lines 394.  

5) Figure 2. What is a "nodular calli"? A callus may be undifferentiated, embryogenic, or shoot-differentiating. The place where the callus tissue is formed is not the main factor in its description.

Response: Please see: Figure 2: 143-146, section 4.2: lines 390-395.

1. Material & Method 4.2: lines 393-396.  

6) There is no statistical analysis in the tables and graphs. There are only error bars or SE values. The text also gives the following: ‘values with different letters are statistically different (P = 0.05), according to Tukey's test’, but there is not a single letter. Are these results statistically insignificant? The graphs do not indicate whether it is SE or SD.

Response: Please see: Table 2: 126-127, Material & Method: 4.7: lines 461-463

7) All graph headings require supplementing the description with statistics, number of repetitions, etc.

Response: Please see: Figure 3-17 with figure legends which indicate that each experiment contains 25 to 30 samples and each experiment with 3- -replication (N=75) lines 126-127; 133-134; 170-171; 176-177; 193-194; 199-200; 233-234; 240-241; 257-258; 263-264; 280-281; 286-287; 300-301; 307-308; 323-324; 329-330; and 461-462.  

8) In Section 4.4 it is stated that there were at least samples?). 4 replicates of 25 to 30 explants (Do the authors mean In Section 4.7 it is stated that each experiment was replicated 3 times and in 4.4 that was at least 4 times? How many individual trials were finally included in the ANOVA?

Response: On average, each experiment has 3 replicates of 25 to 30 explants (sample) with a total of N=75 Samples. Figure 3-17 lines: 126-127; 133-134; 170-171; 176-177; 193-194; 199-200; 233-234; 240-241; 257-258; 263-264; 280-281; 286-287; 300-301; 307-308; 323-324; 329-330; and 461-462.   

9) A uniform way of writing units should be maintained. M, not mol L-1. I understand mg L-1 but moles should be written the same throughout the article.

Response: Please see lines: 400

10)  kg cm-1 is not an SI unit of pressure, please change it.

Response: Please see line 400

11)  How was callus tissue assessed as embryogenic? In section 4.2 you should provide the age of the callus (the results the authors show a callus that is 2 or 3 weeks old). I understand from the methodology that callus was not passaged and conducted as a culture.

Response: Please see the line: 396-397 with reference [34 and 35], lines 398, 399

12)  I also have a question about whether it was necessary to remove the seed coat. Doesn't hemp germinate without it? The entire process could be limited to sterilization and placement on a medium after briefly drying on tissue paper.

Response: Since we have standardized the protocol for seedling transformation by removing the seed coat [Manuscript under preparation] we use the same protocol. please see lines: 378-384

13)  Isn't the USA required permission to carry out genetic modifications? If so, please provide the permission number.

Response: Please see the revised version, not applicable.

14)  In the text are some typos and errors in the text. For example, the second part of the species name should be written with a lowercase letter in Latin. Also, superscripts in units or subscripts in chemical formulas need to be checked.

Response: Please see the revised version, all done.

Reviewer 2 Report

Comments and Suggestions for Authors

The manuscript presents a critical and timely investigation into optimizing Agrobacterium-mediated genetic transformation in industrial hemp, crucial for facilitating research in cannabinoid biosynthesis pathways and advancing functional genomics in this economically important crop. However, the study requires major revisions and further elaboration on several fronts to ensure its scientific rigor and clarity.

The line numbers have been omitted from the manuscript, which makes it difficult to state where revisions should be made.

The manuscript lacks some critical contextual information. It would greatly benefit from a more comprehensive introduction that delineates the importance of genetic transformation in industrial hemp, its current limitations, and the significance of this research within the broader context of hemp biotechnology. Providing a clearer rationale for the parameters chosen and their relevance to transformation efficiency is essential.

The methodology section requires expansion and clarification. Details on the choice of explants, rationale behind the immersion time, the concentrations of silver nitrate, calcium chloride, acetosyringone, and bacterial density are crucial for the readers to understand the experimental design.

The manuscript needs more robust data interpretation. While the results indicate that certain parameters resulted in higher GUS and GFP expression, a deeper analysis of the data, statistical significance, and potential interactions between the optimized parameters is necessary. Including statistical analyses and replicates would strengthen the credibility of the findings.

The discussion section should not only recapitulate the findings but also provide a thorough analysis of the implications of these results in the context of industrial hemp transformation. Furthermore, implications for future research and the practical applications of the optimized transformation method should be clearly stated in the conclusion. The manuscript should conclude with a section dedicated to future directions, including how this optimized transformation system can pave the way for specific research areas such as genome editing, investigating secondary metabolites, and understanding gene functions in cannabinoid biosynthesis pathways.

In summary, while the manuscript touches upon a pivotal area of research, it requires significant revisions to improve its clarity, methodological robustness, and data interpretation. Addressing these concerns will enhance the scientific significance and impact of this study in the field of hemp biotechnology.

Author Response

plants 27141141 Response to reviewer 2 revised version  

The line numbers have been omitted from the manuscript, which makes it difficult to state where revisions should be made.

1) The manuscript lacks some critical contextual information. It would greatly benefit from a more comprehensive introduction that delineates the importance of genetic transformation in industrial hemp, its current limitations, and the significance of this research within the broader context of hemp biotechnology. Providing a clearer rationale for the parameters chosen and their relevance to transformation efficiency is essential.

Response: Please see the revised version 1. Introduction: lines 67-68.

2) The methodology section requires expansion and clarification. Details on the choice of explants, the rationale behind the immersion time, the concentrations of silver nitrate, calcium chloride, acetosyringone, and bacterial density are crucial for the readers to understand the experimental design.

Response: Please see the 2. Material & Methods 4.1 to 4.7: lines; [378 to 383], [396 to 400], [425 to 428], [436 to 437], [461- 463].   

3) The manuscript needs more robust data interpretation. While the results indicate that certain parameters resulted in higher GUS and GFP expression, a deeper analysis of the data, statistical significance, and potential interactions between the optimized parameters is necessary. Including statistical analyses and replicates would strengthen the credibility of the findings.

Response: Please see the 3 Results: 2.1 [103 to 105], Table 1 and Table 2 with figure legend [126-127], 133-134].

4) The discussion section should not only recapitulate the findings but also provide a thorough analysis of the implications of these results in the context of industrial hemp transformation. Furthermore, implications for future research and the practical applications of the optimized transformation method should be clearly stated in the conclusion. The manuscript should conclude with a section dedicated to future directions, including how this optimized transformation system can pave the way for specific research areas such as genome editing, investigating secondary metabolites, and understanding gene functions in cannabinoid biosynthesis pathways.

Response: Please see the revised version

5) Conclusion: lines 465 to 469.

6) In summary, while the manuscript touches upon a pivotal area of research, it requires significant revisions to improve its clarity, methodological robustness, and data interpretation. Addressing these concerns will enhance the scientific significance and impact of this study in the field of hemp biotechnology.

Response: Please see the attached revised version of sections 2-5 for details. 

Reviewer 3 Report

Comments and Suggestions for Authors

The paper suggests various methods for the improvement of industrial hemp genetic transformation, a plant with increasing significance due to its industrial or medicinal use.  

The manuscript is well written, easy to follow, the reference list is elaborate and is included in the text. However, the study design shows several fundamental shortcomings.

I have following remarks and suggestions:

-        The title does not indicate that the study is only about transient expression of the reporter genes, this is somewhat misleading.

-        The plasmid used for the transformation experiments: there is no reference in the text about the pRG8 plasmid. The pCAMBIA1304 mentioned in the description of Figure 1. used as origin contains a gfp:gus reporter fusion. However, this gene does not contain an intron, therefore its expression is not restricted to the eukaryotic plant cells. The positive signal might be detected from bacterial expression as well. Agrobacteria are very persistent, not easy to kill off and strive on co-inoculated tissue despite of antibiotic treatment.  

-        Line 114: Table 1. The description should indicate that the table summarizes transient expression values in different explants

Figures with the different parameters are presented in a reasonable form, but the axes are not defined. The authors describe in the materials and methods (lines 427-434), that they used different explant types for the measurement of the parameter effects (leaf, cotyledon, hypocotyl, root, nodal part, and leaf-derived embryogenic callus), but the different explant types are not represented in the figures. (Table 1. shows a summary about reporter gene expression in the different explants, but does not reflect the parameters).   

-        I also have a problem with understanding the figures showing the effect of the age of the explant and the wounding methods. Here, the percentage/number of spots per leaves are depicted, but it is not obvious if an absolute or relative leaf are is taken into consideration. To me it seems from the pictures that the leaves have different sizes, which might have an influence on the total area of the observed reporter gene expression.

-        Lines 202-205 and lines 263-264: how did you come to this conclusion?

-        Line343-345: “In this study needle wounding, screen wounding, and gene gun wounding were tested, and found that leaf tissue wounded with gene gun had higher relative GUS and 344 GFP expression compared to others.” – gene gun wounding has not been mentioned previously, especially not that it was the most efficient method…

-        Line361: “These results are consistent with studies conducted in Phalaenopsis Violacea and Dendrobium”- reference [25] is about banana

-        Line368: “The C. Sativa variety ‘Joey’ seeds were obtained from local sources.” – the variety should be described in more detail. Is it a true-breeding line? According to the introduction hemp is highly heterogenic (line36), which might strongly influence genetic transformation efficiencies. Were then the explant samples always taken from the same individuals to reduce heterogenic nature of the samples?

-        Line401: there is no information or a reference about plasmid pRG8

-        Line405-406: Which agrobacterium strain was used? “Agrobacterium strain LBA4404 carrying pRG8 was used for transformation.” Vs. “Explants were transformed with the binary vector pRG8 derived from pCAMBIA1304 via Agrobacterium tumefaciens (strain GV3101).”- lines 130-131

Typos:

-        Redundant “.” at section titles, e.g. line 89:  “.Results”,

                       line 90: “. Transient Expression Efficiency Based on Different Tissue and Age of Explants:”

                       line170: “. Effect of Bacterial Cell Density and Bacterial Immersion Period on Transient Expression Efficiency:” etc.   

-        Abbreviations: “day” - “d” or “week” – “wk” are randomly used in the text, should be day(s) and week(s)

-        Line457: X-Gluc instead of X-Glue

Author Response

  1. plants 27141141 Response to reviewer 3 revised version

Response to Reviewer 3 Comments

The manuscript is well-written and easy to follow, the reference list is elaborate and is included in the text. However, the study design shows several fundamental shortcomings.

I have the following remarks and suggestions:

1) The title does not indicate that the study is only about transient expression of the reporter genes, this is somewhat misleading.

Response: Please see the revised version MS title change. Line 2-3,5.

2) The plasmid used for the transformation experiments: there is no reference in the text about the pRG8 plasmid. The pCAMBIA1304 mentioned in the description of Figure 1. used as the origin contains a GFP:gus reporter fusion. However, this gene does not contain an intron, therefore its expression is not restricted to the eukaryotic plant cells. The positive signal might be detected from bacterial expression as well. Agrobacteria are very persistent, not easy to kill off, and strive on co-inoculated tissue despite antibiotic treatment.

Response: Please see the revised version section 4.3: Material & Methods: lines 405-409. 

3) Line 114: Table 1. The description should indicate that the table summarizes transient expression values in different explants.

Response: Please see the revised version of Lines: 123-129.

4) Figures with the different parameters are presented in a reasonable form, but the axes are not defined. The authors describe in the materials and methods (lines 427-434), that they used different explant types for the measurement of the parameter effects (leaf, cotyledon, hypocotyl, root, nodal part, and leaf-derived embryogenic callus), but the different explant types are not represented in the figures. (Table 1. shows a summary of reporter gene expression in the different explants but does not reflect the parameters).

Response: Please see the revised version with Tables 1 and Table 2: Lines: 124-129; and Figures 3- 17: lines 126-127; 133-134; 170-171; 176-177; 193-194; 199-200; 233-234; 240-241; 257-258; 263-264; 280-281; 286-287; 300-301; 307-308; 323-324; 329-330; and 461-462.  

5) I also have a problem with understanding the figures showing the effect of the age of the explant and the wounding methods. Here, the percentage/number of spots per leaf is depicted, but it is not obvious if an absolute or relative leaf is taken into consideration. To me, it seems from the pictures that the leaves have different sizes, which might have an influence on the total area of the observed reporter gene expression.

Response: Please see the revised version with Figure 3: lines 166-177. 

6) Lines 202-205 and lines 263-264: how did you come to this conclusion?

Response: Please see the revised version of line; 210 with the cited reference.

7) Line 343-345: “In this study needle wounding, screen wounding, and gene gun wounding were tested, and found that leaf tissue wounded with gene gun had higher relative GUS and 344 GFP expression compared to others.” – Gene gun wounding has not been mentioned previously, especially not that it was the most efficient method.

Response: Please see the revised version 2.3: lines 201 to 213; and Figure 7 [215 to 221], Figure 8 [229 to 234] Figure 9 [236 to 241].  

8) Line 361: “These results are consistent with studies conducted in Phalaenopsis Violacea and Dendrobium”- reference [25] is about bananas.

Response: Please see the revised version line 371.  

9) Line 368: “The C. Sativa variety ‘Joey’ seeds were obtained from local sources.” – the variety should be described in more detail. Is it a true-breeding line? According to the introduction, hemp is highly heterogenic (line 36), which might strongly influence genetic transformation efficiencies. Were the explant samples always taken from the same individuals to reduce the heterogenic nature of the samples?

Response: Please see the revised version of lines 378-380.  

10) Line 401: there is no information or a reference about plasmid pRG8.

Response: Please see the revised version: lines 405 to 410.  

11) Line 405-406: Which agrobacterium strain was used? “Agrobacterium strain LBA4404 carrying pRG8 was used for transformation.” Vs. “Explants were transformed with the binary vector pRG8 derived from pCAMBIA1304 via Agrobacterium tumefaciens (strain GV3101).”- lines 130-131

Response: Please see the revised version 1. Introduction: lines 137-138 and 405-406.

12) Typos: Redundant “.” at section titles, e.g. line 89: .Results”,

line 90: “. Transient Expression Efficiency Based on Different Tissue and Age of Explants:”

Response: line: 99

13) line170: “. Effect of Bacterial Cell Density and Bacterial Immersion Period on Transient Expression Efficiency:” etc.

Response: line: 178

14) Abbreviations: “day” - “d” or “week” – “wk” are randomly used in the text, should be day(s) and week(s)

Response: line: days(s) and week(s) done in the revised version.

15) Line457: X-Gluc instead of X-Glue

Response: line: 450

Round 2

Reviewer 2 Report

Comments and Suggestions for Authors

All the comments have been addressed. The current version of the manuscript can be published in Plants.

Reviewer 3 Report

Comments and Suggestions for Authors

Although the authors improved the text, they still have not addressed the major concerns about the manuscript. I recommend rejection.

I have following remarks to the responses:

(The lines do not correspond with the given reference in the text. It is difficult to make out to which lines the authors refer.)

2.     The plasmid used for the transformation experiments: there is no reference in the text about the pRG8 plasmid. The pCAMBIA1304 mentioned in the description of Figure 1. used as the origin contains a GFP:gus reporter fusion. However, this gene does not contain an intron, therefore its expression is not restricted to the eukaryotic plant cells. The positive signal might be detected from bacterial expression as well. Agrobacteria are very persistent, not easy to kill off, and strive on co-inoculated tissue despite antibiotic treatment. 

Response: Please see the revised version section 4.3: Material & Methods: lines 405-409.

The CaMV 35S promoter is known to be active in bacteria. Since there is no intron integrated in the protein coding region the reporter gene expression might result from Agrobacteria instead of the plant cells. The authors did not response to this issue, which, I believe, is a fundamental weakness of the experimental setup.

4.     Figures with the different parameters are presented in a reasonable form, but the axes are not defined. The authors describe in the materials and methods (lines 427-434), that they used different explant types for the measurement of the parameter effects (leaf, cotyledon, hypocotyl, root, nodal part, and leaf-derived embryogenic callus), but the different explant types are not represented in the figures. (Table 1. shows a summary of reporter gene expression in the different explants but does not reflect the parameters).

Response: Please see the revised version with Tables 1 and Table 2: Lines: 124-129; and Figures 3- 17: lines 126-127; 133-134; 170-171; 176-177; 193-194; 199-200; 233-234; 240-241; 257-258; 263-264; 280-281; 286-287; 300-301; 307-308; 323-324; 329-330; and 461-462.  

I do not see a significant improvement in the presentation of the data. Still the different explant types are not represented in the figures, only in tables 1 and 2.

5.     I also have a problem with understanding the figures showing the effect of the age of the explant and the wounding methods. Here, the percentage/number of spots per leaf is depicted, but it is not obvious if an absolute or relative leaf is taken into consideration. To me, it seems from the pictures that the leaves have different sizes, which might have an influence on the total area of the observed reporter gene expression.

Response: Please see the revised version with Figure 3: lines 166-177. 

 Only the absolute number of expression spots were taken into consideration, but the evaluation did not consider that the different leaves have different surface area. The authors also mention that the hemp donor material is highly heterogenic. This might strongly influence transformation efficiency if the samples are taken from different individuals. Still no information about this.

9.     Line 368: “The C. Sativa variety ‘Joey’ seeds were obtained from local sources.” – the variety should be described in more detail. Is it a true-breeding line? According to the introduction, hemp is highly heterogenic (line 36), which might strongly influence genetic transformation efficiencies. Were the explant samples always taken from the same individuals to reduce the heterogenic nature of the samples?

Response: Please see the revised version of lines 378-380. 

The respective lines are not responding to this issue. The only information I found is that the authors added information about the source of the hemp seeds, but I found no answers to my other questions.